

# Multimodal dataset for sensor fusion in fall detection

Carla Taramasco[1,2], Miguel Pineiro[1], Pablo Ormeño-Arriagada[3], Diego Robles[4,5] and David Araya[1]

[1] Facultad de Ingeniería, Universidad Andrés Bello, Vina del Mar, Valparaíso, Chile
[2] Millenium Núcleo of SocioMedicine (SocioMed), Universidad Mayor, Santiago, Region Metropolitana, Chile
[3] Escuela de Ingenieria y Negocios, Universidad de Viña del Mar, Vina del Mar, Valparaíso, Chile
[4] Escuela de Ingeniería Civil Informática, Universidad de Valparaíso, Valparaiso, Valparaíso, Chile
[5] Escuela de Kinesiología, Facultad de Salud y Odontología, Universidad Diego Portales, Santiago, Region Metropolitana, Chile

Corresponding author
Miguel Pineiro,
miguel.andres.pineiro.feick@gmail.com

## ABSTRACT

The necessity for effective automatic fall detection mechanisms in older adults is driven by the growing demographic of elderly individuals who are at substantial health risk from falls, particularly when residing alone. Despite the existence of numerous fall detection systems (FDSs) that utilize machine learning and predictive modeling, accurately distinguishing between everyday activities and genuine falls continues to pose significant challenges, exacerbated by the varied nature of residential settings. Adaptable solutions are essential to cater to the diverse conditions under which falls occur. In this context, sensor fusion emerges as a promising solution, harnessing the unique physical properties of falls. The success of developing effective detection algorithms is dependent on the availability of comprehensive datasets that integrate data from multiple synchronized sensors. Our research introduces a novel multisensor dataset designed to support the creation and evaluation of advanced multisensor fall detection algorithms. This dataset was compiled from simulations of ten different fall types by ten participants, ensuring a wide array of scenarios. Data were collected using four types of sensors: a mobile phone equipped with a single-channel, three-dimensional accelerometer; a far infrared (FIR) thermal camera; an $8\times8$ LIDAR; and a 60–64 GHz radar. These sensors were selected for their combined effectiveness in capturing detailed aspects of fall events while mitigating privacy issues linked to visual recordings. Characterization of the dataset was undertaken using two key metrics: the instantaneous norm of the signal and the temporal difference between consecutive frames. This analysis highlights the distinct variations between fall and non-fall events across different sensors and signal characteristics. Through the provision of this dataset, our objective is to facilitate the development of sensor fusion algorithms that surpass the accuracy and reliability of traditional single-sensor FDSs.

## INTRODUCTION

The global increase in elderly populations presents significant challenges for home care systems and the prevention of accidents, particularly among seniors who live independently (*Sanderson, Scherbov & Gerland, 2017*). A prevalent risk for this demographic is the incidence of falls, which remains a critical concern (*Kalache et al., 2007*). According to the World Health Organization, approximately 28–35% of people over the age of 65 fall each year, and this figure increases to 32–42% for those aged 70 or older (*Kalache et al., 2007*). The likelihood of falling increases with age and as individuals become more frail.

The prevalence of falls among the elderly is a significant concern due to the high risks of injury and the impact on quality of life. According to recent studies, falls are a leading cause of injury-related deaths in older adults, highlighting the urgent need for effective fall detection systems (FDSs).

Crucially, research involving 110 individuals aged over 90 years showed that only about half of those who fell could get up without assistance (*Fleming & Brayne, 2008*). Falls can lead to various adverse health outcomes ranging from minor injuries such as fractures (*Melton et al., 2010*) and abrasions to severe complications like dehydration, hypothermia, pneumonia (*Fleming & Brayne, 2008*), and more severe conditions such as internal infections, bleeding, cellulitis, ulcers, chest pain, fainting, heart attacks, and potentially death (*Tinetti, Liu & Claus, 1993*). The psychological impact is also significant, as many seniors develop a fear of falling again, severely limiting their mobility and daily activities (*Ambrose, Paul & Hausdorff, 2013*).

However, current classifications for aging-related diseases are fragmented, limiting effective diagnosis and intervention for fall risks. A comprehensive framework encompassing cellular dysfunction and tissue senescence could improve diagnostic precision and preventative care. Such a system would enable better health outcomes for elderly populations by supporting tailored interventions for aging-related conditions (*Calimport et al., 2019*).

In response to these issues, the last decade has seen an increased focus on developing and enhancing FDSs that not only detect falls but also provide early assistance and help prevent falls among the elderly during their routine activities (*Forbes, Massie & Craw, 2020*; *Fischinger et al., 2016*). FDSs have also been incorporated into care robots, enabling them not only to detect falls but also to respond to them (*Fischinger et al., 2016*; *Wei et al., 2024*; *Elwaly, Abdellatif & El-Shaer, 2024*). FDSs typically use computational algorithms based on predictive analytics or machine learning techniques and require comprehensive training datasets to accurately differentiate between actual falls and normal daily activities like walking or sitting. Fall detection technology categorizes into three primary types: visual, ambient, and wearable sensors. Wearable sensors have the disadvantage of being invasive, requiring the user to continuously wear a device; if forgotten or uncharged, monitoring fails. Ambient systems are highly sensitive to environmental changes, such as furniture movement, which demands frequent adjustments to maintain accuracy. Visual sensors, while effective, can raise privacy concerns and may perform poorly in

low-light conditions. Visual sensors encompass technologies such as visible spectrum cameras (*Rajabi & Nahvi, 2015*; *El Kaid, Baïna & Baïna, 2019*; *Shu & Shu, 2021*), infrared cameras (*Taramasco et al., 2018*; *Mastorakis & Makris, 2014*), and Microsoft Kinect (*Yajai et al., 2015*; *Kalinga et al., 2020*). Ambient sensors include options like radio frequency (RF) sensors (*Ji, Xie & Li, 2023*; *Mager, Patwari & Bocca, 2013*), Doppler radars (*Yoshino, Moshnyaga & Hashimoto, 2019*; *Liang et al., 2012*), and LIDAR sensors (*Frovik, Malekzai & ovsthus, 2021*; *Bouazizi, Ye & Ohtsuki, 2021*). Wearable technologies involve devices such as accelerometers, gyroscopes, magnetometers, barometers, and inertial measurement units (*Khojasteh et al., 2018*; *Mahmud & Sirat, 2015*; *Sucerquia, López & Vargas-Bonilla, 2018*).

Despite strides in technological and algorithmic development, overcoming the complexities of FDS remains fraught with challenges (*Thakur & Han, 2022*; *Orejel Bustos et al., 2023*). These challenges are highlighted by ongoing issues like high false positive rates and difficulties in distinguishing actual falls from routine activities.

The core of these difficulties lies in the real-world complexities of residential environments where falls occur. Varied home layouts, furniture placements, and individual movement habits create numerous scenarios that can lead to falls. To effectively mitigate these challenges, it is crucial to advance the development of FDSs that are adept at managing these environmental complexities (*Igual, Medrano & Plaza, 2013*; *Xu, Zhou & Zhu, 2018*). Sensor fusion algorithms offer an approach to overcoming the challenges of accurate fall detection within FDSs (*Cagnoni et al., 2009*; *Wang, Ellul & Azzopardi, 2020*). By combining data from various sensors, these algorithms not only enhance detection accuracy but also significantly reduce both false positives and false negatives, which are critical for reliable fall detection. For example, integrating data from cameras and accelerometers (*Ozcan, Velipasalar & Varshney, 2016*), along with data from doppler radar and infrared sensors (*Liu et al., 2014*), boosts the reliability of detections. A fundamental requirement for these algorithms is the availability of data from multiple sensors.

Despite the exploration of various multimodal approaches to fall detection in existing studies, most datasets lack a wide range of privacy-conscious sensors. To the best of the authors' knowledge, current multisensor databases do not include radar, low-cost LIDAR, and thermal imaging. This article distinguishes itself by presenting a comprehensive multisensor dataset that incorporates these sensors along with an accelerometer, effectively addressing these gaps. Our main contribution lies in creating a privacy-preserving dataset that facilitates future advancements in sensor fusion algorithms and real-world fall detection applications.

## RELATED WORK

The field of fall detection research has seen significant growth in multisensorial datasets, emphasizing the importance of integrating various sensor types for comprehensive monitoring. Systematic reviews highlight the benefits of multisensory data, enabling researchers to collect detailed information that enhances the accuracy and reliability of fall detection systems.

The URFall dataset (*Kwolek & Kepski, 2014*) integrates synchronized inertial data from an IMU, including accelerometer and gyroscope readings, and visual data from a Kinect sensor, which provides both RGB and depth information. It includes data from five subjects performing 30 simulated falls from standing and sitting positions, as well as daily activities.

The CMDFALL dataset (*Tran et al., 2018*) was developed to support human fall analysis by integrating multimodal data from various sensors, including seven Kinect cameras providing RGB, depth, and two accelerometers. The dataset features activities from 50 subjects, each performing 20 tasks that include eight types of falls and 12 common daily activities.

The UP-Fall Detection and Activity Recognition dataset (*Martínez-Villaseñor et al., 2019*) comprises multimodal data gathered from four inertial measurement units (IMUs) positioned on the neck, waist, thigh, and wrist, an electroencephalogram (EEG) helmet, and four ambient infrared presence sensors, capturing accelerometer, gyroscope, EEG, and infrared presence/absence signals. Seventeen volunteers, aged 18 to 24, each performed six daily activities and five types of simulated falls, with each activity repeated three times per subject.

The KFall dataset (*Yu, Jang & Xiong, 2021*) comprises motion data collected from 32 male participants with an average age of $24.9 \pm 3.7$ years, performing 21 types of daily activities and 15 simulated falls. A nine-axis inertial sensor on the lower back recorded acceleration, angular velocity, and orientation at 100 Hz, with synchronized video at 90 Hz.

There are relatively few public databases that simultaneously provide data from both wearable and contextual sensors. Multisensory datasets leverage these varied modalities to capture a comprehensive range of fall-related information. This approach addresses the limitations of single-modality datasets, enabling a more nuanced understanding of fall events and supporting the development of more accurate and effective detection systems.

In comparison to the datasets presented in Table 1, our dataset includes a diverse array of multimodal sensors, each of distinct nature, that simultaneously capture various aspects of falls across different falling scenarios. Additionally, this dataset distinctively avoids the use of RGB cameras to ensure the privacy of individuals. Furthermore, by utilizing low-cost sensors, we make our solution more accessible for broader research and practical deployment.

## METHODS

### Ethical approval and study procedures

This study was conducted with the approval of the Ethics Committee of Universidad Andrés Bello, under Approval Act number 032/2023. The committee reviewed and approved all proposed ethical and methodological aspects, ensuring the protection and respect for the dignity and rights of the participants. The research focused on data collection for the "QUIDA dataset", aimed at enhancing FDSs, specifically designed for the elderly population.

Participants were fully informed about the study objectives, involved procedures, potential benefits, and associated risks of their participation. All activities were conducted

**Table 1 Public datasets on falls that incorporate both wearable and contextual sensors.** Columns include the dataset name, reference, total recorded falls, types of falls, number of participants, age range, and sensor modalities used. In the sensor modality column: *acc* represents accelerometer data, *gyro* stands for gyroscopic data, *RGB* indicates RGB camera video, depth refers to depth map video, *EEG* denotes electroencephalographic data, *IR* refers to infrared presence sensor data, and *mag* indicates magnetometer data.

| Dataset | Reference | Falls | Types | Subjects | Ages | Modalities |
|---------|-----------|-------|-------|----------|------|------------|
| URFall | *Kwolek & Kepski (2014)* | 30 | 4 | 5 | 26– | acc, gyro, RGB, depth |
| CMDFall | *Tran et al. (2018)* | 400 | 8 | 50 | 21–40 | acc, RGB, depth |
| UPFall | *Martínez-Villaseñor, Ponce & Espinosa-Loera (2018)* | 255 | 5 | 17 | 18–24 | acc, gyro, EEG, IR |
| KFall | *Yu, Jang & Xiong (2021)* | 2346 | 15 | 32 | 24.9 ± 3.7 | acc, gyro, mag, RGB |

in a controlled and safe environment, using gymnastics mats and other safety measures to minimize any risk of injury. Participation was voluntary, and all participants provided written informed consent before participating in the study.

The confidentiality and anonymity of the collected data were paramount. In line with ethical practices recommended by the committee, no digital records of sensitive participant data were made. Only necessary data for the study objectives were encoded and stored, and only authorized personnel had access to this information. This approach ensures the integrity and privacy of the information, complying with the ethical standards required.

## Data capture

The data was captured using four different sensors in a room with gymnastics mats placed in the middle and a desktop to the side. The center of the mats is the designated fall area and people move along the length of the mats for the experiments.

The first sensor is the accelerometer of a Samsung Galaxy A14 smartphone, securely placed on the participant's torso between the T5 and T10 vertebral levels using an adjustable harness. This anatomical location was chosen due to its proximity to the body's center of mass, providing a more representative measure of trunk balance and control during movement. The smartphone integrates an STMicroelectronics LSM6DSL 6-DoF inertial measurement unit (IMU), whose accelerometer has a maximum acceleration range of $\pm16$ g with a resolution of 0.488 mg. Data was recorded at a sampling frequency of 62.5 Hz (16 ms period), which is well-suited for capturing falls, as their primary frequency components typically lie within the 2–3.5 Hz range (*Huynh & Tran, 2021*). This sampling rate also exceeds the recommended minimum of 40 Hz for accelerometers used in physical activity monitoring (*Meng & Kim, 2011*).

Mounted on the ceiling approximately 2.7 m above the ground and directly above the mats, an $8 \times 8$ STMicroelectronics VL53L5CX LIDAR was installed. It was programmed to operate at 10 Hz (100 ms period), with an approximate field of view of $2 \times 2$ meters centered on the fall area. About 20 cm from the LIDAR sensor, a Texas Instruments IWR6843AOP 60–64 GHz radar was installed. It operated at 8.3 Hz (120 ms period) and had an approximate field of view of $4 \times 4$ meters. Given its broad area of vision, the radar was off center from the fall area. Finally, a Melexis MLX90640 $32 \times 24$ far infrared (FIR) thermal camera was strategically mounted on the desktop 1.5 m from and centered on the fall area, at 1 meter above ground and programmed to operate at 2.6 Hz (384 ms period).

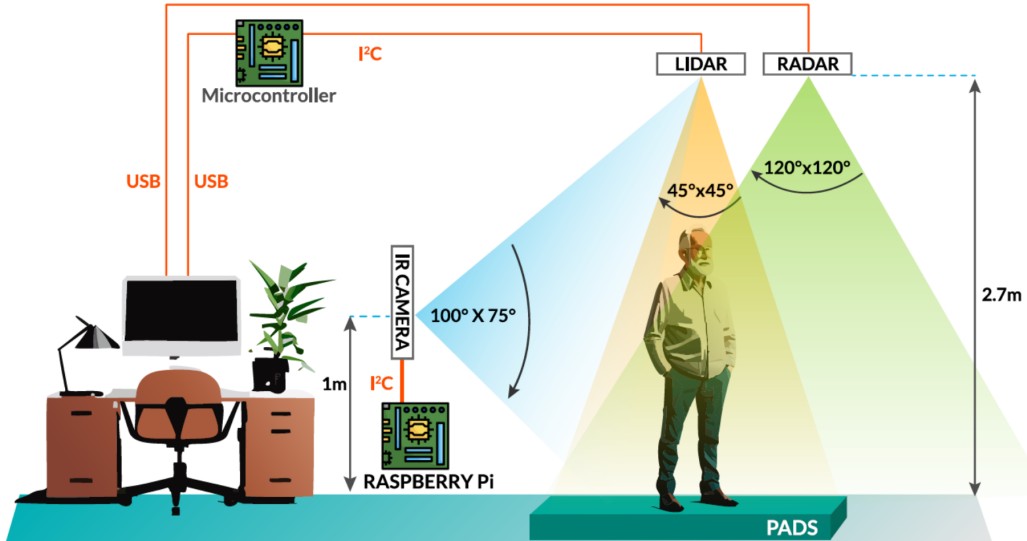

**Figure 1  System setup.**

The FIR camera interfaced with a Raspberry Pi 4 *via* I$^2$C, the LIDAR sensor was attached to an ESP32 microcontroller *via* I$^2$C, which then connected to a computer through USB, and the radar was directly connect to a computer through USB, as shown in Fig. 1.

Figure 2 illustrates the block diagram of the multisensor system, depicting the arrangement of the various components including sensors, a Raspberry Pi, and other processing units. The diagram outlines the data flow and connectivity between the devices, such as the harness with a mobile phone, LIDAR, radar, and a far infrared thermal camera, all integrated into a room designed for fall detection experiments.

Throughout the experiments, continuous data recording of all sensors at the same time took place, with the exception of occasional interruptions caused by cable unplugging incidents during the data collection process. The mobile phone's accelerometer was configured to record continuous blocks of 12 s for each fall and during walking period of the protocol. All sensors, with the accelerometer being the sole exception, employed custom software for the capture and recording of sensor data. Figure 3 shows a sample of the data captured by these sensors.

It should be noted that the sensors used in this study (LIDAR, thermal cameras, accelerometers and radar) intrinsically favor privacy, since the information they capture does not allow personal identification. In addition, the data reported are presented anonymously, protecting the identity of the participants.

## Dataset generation

In order to create the dataset, each participant was required to perform a series of tasks. The study involved a phase of walking without falling and simulating 10 different types of falls. The activities were sequenced as follows:

1. Walking without falling

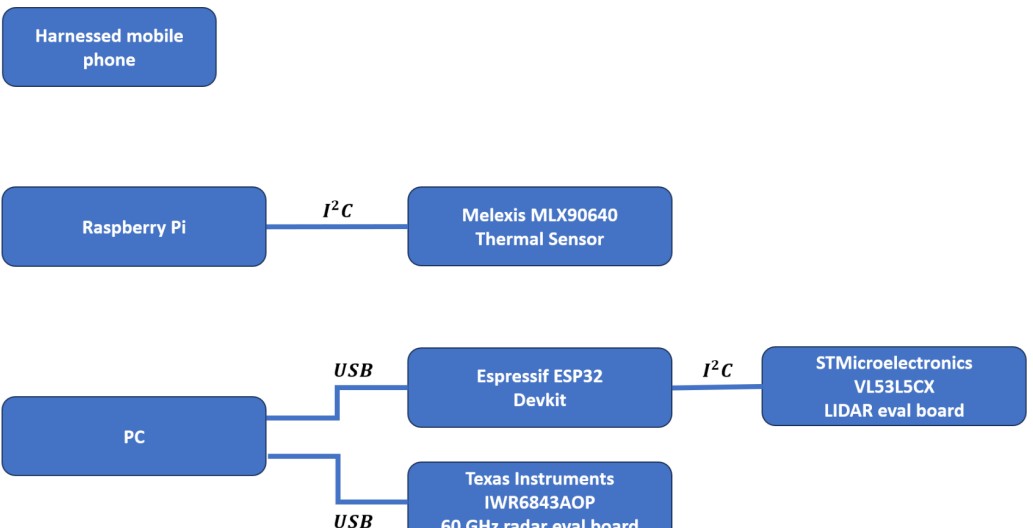

**Figure 2    Block diagram of multisensor system setup.**

2.  Falling backward while walking backward
3.  Falling forward caused by tripping
4.  Falling caused by fainting
5.  Falling backward while attempting to sit down
6.  Falling backward with straight legs
7.  Falling forward with straight legs
8.  Falling forward with knee flexion
9.  Falling backward with knee flexion
10.  Lateral falling with straight legs
11.  Lateral falling with knee flexion.

Except for the mobile phone accelerometer, custom Python programs were used to capture data from the various sensors. All the captured data included a timestamp which was then used to synchronize the time series of all the sensors.

## Dataset description

The dataset supporting this study is publicly accessible on the Open Science Framework (OSF) platform at https://doi.org/10.17605/OSF.IO/YJGDV. It is organized with individual directories for each subjects, containing four CSV files corresponding to the different sensors. The mean and standard deviation for the age, weight, height, and Body Mass Index (BMI) of the participants are presented in Table 2. In each sensor's CSV file, the first column presents the timestamp in Unix time, with subsequent columns providing sensor-specific data.

Columns 2, 3, and 4 of the accelerometer CSV contain acceleration along the $X$-axis (left–right), $Y$-axis (craniocaudal), and $Z$-axis (anteroposterior), respectively, measured in meters per second squared (m/s$^2$).

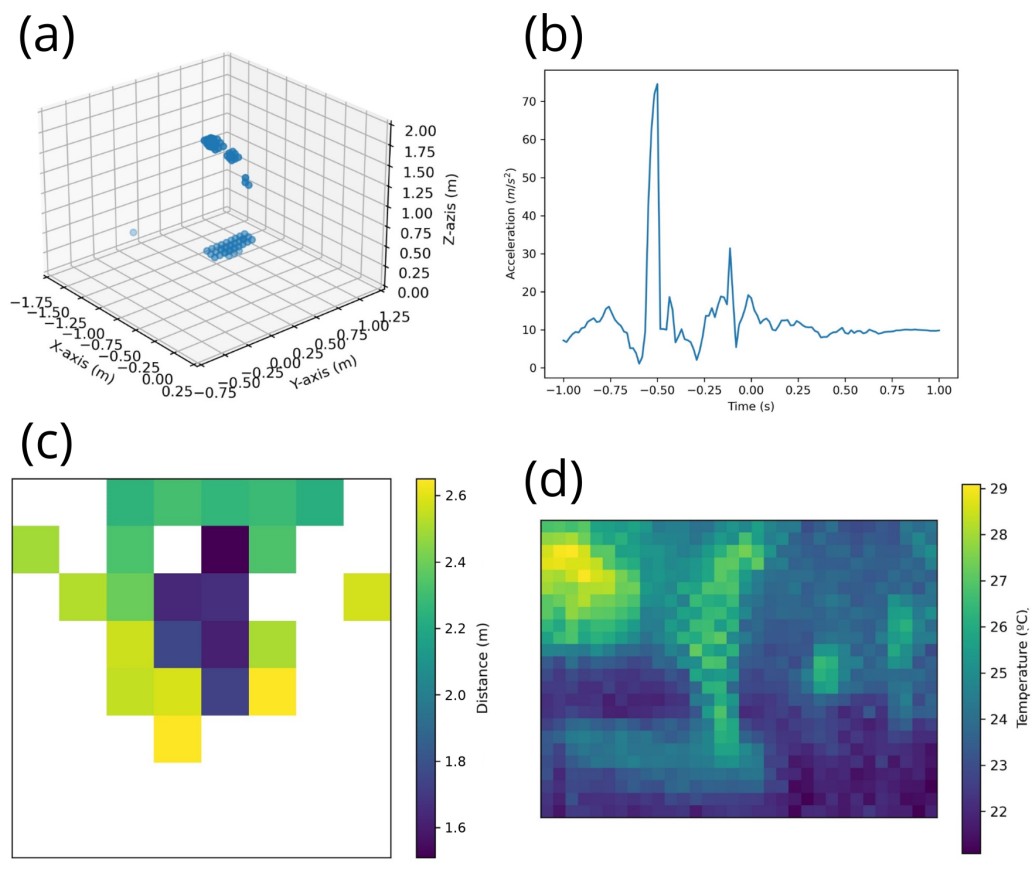

**Figure 3** (A) Radar example illustrating the 3D positions of detected points of a standing subject. (B) Accelerometer example showing the acceleration vector magnitude during a fall event. (C) LIDAR example depicting a standing subject below the sensor. (D) Infrared camera example displaying a standing subject.

**Table 2** Statistics of the subjects.

| Characteristic | Male | Female |
| --- | --- | --- |
| Subjects (n) | 8 | 2 |
| Age (years) | 24.5 ± 3.51 | 25.5 ± 2.12 |
| Weight (kg) | 78.62 ± 7.11 | 74 ± 2.70 |
| Height (m) | 1.78 ± 0.05 | 1.63 ± 0.09 |
| Body Mass Index (kg/m$^2$) | 24.56 ± 1.23 | 27.75 ± 7.28 |

The FIR camera's CSV file contains data in columns 2 to 769, representing the temperature in degrees Celsius (°C) for each pixel in a $32 \times 24$ matrix.

In the LIDAR CSV, columns 2 to 65 detail the ambient light received by the Single-photon avalanche diode (SPAD) array, quantified in kilo-counts per second per SPAD (kcps/SPAD). Columns 66 through 129 in the dataset represent the count of targets identified for each matrix element. In these columns, a value of 1 denotes a successful distance measurement at the respective matrix element, whereas a value of 0 indicates an

inability to measure the distance. Columns 130 to 193 show the number of SPADs active for the current measurement. Columns 194 to 257 represent the photon count during the vertical-cavity surface-emitting laser (VCSEL) pulse, also in kcps/SPAD. The standard deviation of the measured distance in meters is specified in columns 258 to 321. Columns 322 to 385 provide the distance measurements in meters for the matrix elements, and columns 386 to 449 describe the status of each matrix element, with more details available in the UM2884 document by STMicroelectronics. Finally, columns 514 to 577 provide the reflectance percentage of the object.

The radar CSV file uses a variable number of columns on each row depending on the number of points detected. Specifically, the number of columns equals $1+7n$ for $n$ detected points. Every block of seven consecutive columns represents a single point and their rows indicate the (x,y,z) position (m), Doppler (m/s), SNR (dB), noise (dB) and tracking ID.

The file named 'Falls.csv', located in the root directory, records the timings of falls for the different participants. The data is organized such that each column corresponds to a different subject, arranged sequentially. The first column pertains to the first subject, the second column to the second subject, and this pattern continues across columns. Additionally, each row in the file represents a separate instance of a fall.

## Dataset segmentation

To separate fall and no-fall events within the dataset, the following procedure was used to segment each recording into discrete frames. This segmentation was based on manual annotations marking the exact instances of falls captured in the dataset. Following each annotated fall, a predefined interval determined by user-defined parameters, *e.g.*, 2 s, defines a window both 2 s before and 2 s after the annotated fall event. This window size is chosen to encompass the entire fall event, ensuring that all relevant data points within the segment are captured.

For cases not classified as falls, segments of equal width were defined using measurements outside the segments labeled as falls, maintaining a balance between fall and no-fall samples at a ratio of 1:N, in this case $N = 3$. This segmentation strategy not only guarantees the representation of both event types within the dataset, but also allows the synchronization of events across different sensor data streams, despite their different sampling rates.

The synchronization of the four sensors was ensured by performing a specific action that generated a clear, simultaneous peak across all signals, achieved through a sudden movement accompanied by arm flapping. The RGB camera was also included in this synchronization process. The exact moment of the fall was determined by analyzing the visual recording, allowing for precise alignment of the sensor data with the observed fall event. This approach provided a consistent reference point for data analysis across all modalities.

Thus, each segment, whether representing a fall or a no-fall event, is synchronized to represent the same event across all sensor modalities, albeit with a different number of data points due to the different sampling frequencies of each sensor. This approach ensures a comprehensive and consistent representation of each event, facilitating subsequent analysis
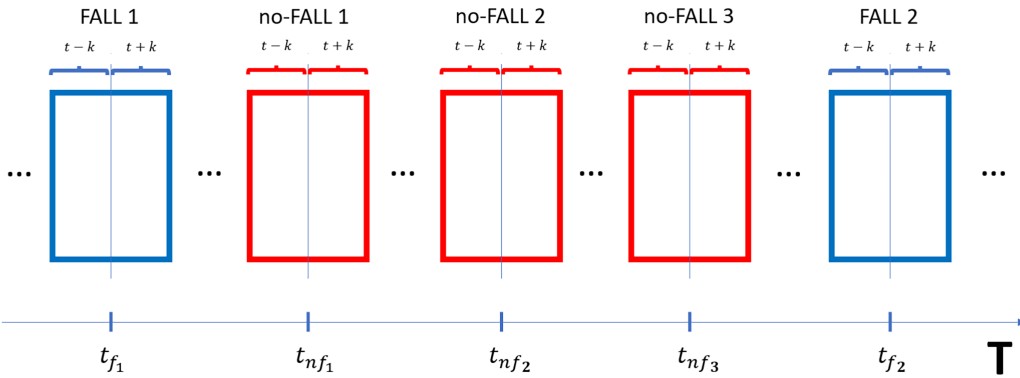

**Figure 4  Illustration of the segmentation tool employed to divide the dataset into Fall and No-Fall segments.**

and classification based on these segmented data sets. To visualize the segmentation process described, refer to the schematic illustration provided in Fig. 4.

## Feature extraction

For characterizing the dataset, our aim is to distinguish between fall and non-fall events using simple yet effective features. This approach helps to emphasize the fundamental differences between these event categories, facilitating their future identification and analysis.

To characterize the captured sensor data from matrices such as those from LIDAR, radar, and thermal cameras, the Frobenius norm (or L2 norm) is employed in two distinct ways: instantaneously and as a difference between consecutive frames. The Frobenius norm, a measure of a matrix's magnitude, is calculated by squaring each element of the matrix, summing all these squared values across both axes, and then taking the square root of this sum (*Amir et al., 2024*; *Jefiza et al., 2017*).

The mathematical formulations for instantaneous metric is as follows:

$$\text{Instantaneous Matrix Norm} = \sqrt{\sum_{i=1}^{m}\sum_{j=1}^{n}|a_{ij,t}|^2} \tag{1}$$

where A denote the signal matrix $A_t$ at time $t$, where $x$ and $y$ represent the rows and columns of the matrix, respectively. This norm reflects the instantaneous power or intensity of the sensor data.

The mathematical formulations for difference between the norms of consecutive frames:

$$\text{Matrix Norm Difference} = \sqrt{\sum_{i=1}^{m}\sum_{j=1}^{n}|a_{ij,t} - a_{ij,t-1}|^2} \tag{2}$$

For the three signals from the accelerometer, the Euclidean norm (or L2 norm) is utilized to analyze the signals' aggregate intensity at any given instant as well as the intensity changes between consecutive readings.

**Table 3  Summary of the different types of falls, alongside corresponding measurements obtained from various sensors.** Each cell displays the mean value, with the standard deviation provided in parentheses.

| Fall type | radar (m) | accel. (m/s²) | thermal (°C) | LIDAR (m) |
|---|---|---|---|---|
| Backward while walking backward | 1.114 (0.941) | 10.350 (2.356) | 13.843 (2.541) | 0.511 (0.358) |
| Forward caused by tripping | 1.292 (1.090) | 10.568 (3.353) | 14.774 (2.772) | 0.392 (0.253) |
| Caused by fainting | 1.398 (0.980) | 10.533 (2.467) | 15.822 (2.861) | 0.381 (0.275) |
| Backward while attempting to sit down | 1.422 (0.997) | 10.323 (1.902) | 14.925 (2.517) | 0.441 (0.289) |
| Backward with straight legs | 1.307 (0.958) | 10.617 (2.571) | 14.031 (2.905) | 0.332 (0.213) |
| Forward with straight legs | 1.261 (0.848) | 10.640 (2.590) | 14.031 (2.519) | 0.444 (0.299) |
| Forward with knee flexion | 1.085 (0.846) | 10.359 (2.209) | 13.243 (1.985) | 0.375 (0.256) |
| Backward with knee flexion | 1.371 (0.980) | 10.772 (3.576) | 14.881 (3.744) | 0.542 (0.373) |
| Lateral falling with straight legs | 1.527 (1.089) | 10.528 (2.526) | 14.142 (2.640) | 0.542 (0.341) |
| Lateral falling with knee flexion | 1.407 (1.085) | 10.687 (3.071) | 14.977 (3.077) | 0.440 (0.278) |
| No falls | 1.236 (1.434) | 10.144 (2.934) | 13.474 (5.425) | 0.399 (0.476) |

The mathematical formulation for the instantaneous metric is expressed as follows:

$$\text{Vector Norm} = \sqrt{x_t^2 + y_t^2 + z_t^2}. \tag{3}$$

This equation calculates the Euclidean norm for the accelerometer data at time $t$, where $x_t$, $y_t$, and $z_t$ denote the acceleration values along the X, Y, and Z axes, respectively. This measure reflects the overall intensity or "power" of the motion captured by the accelerometer at that specific moment.

To capture the dynamic nature of the data by analyzing changes between consecutive readings, the difference in norms between two successive frames is computed as:

$$\text{Vector Difference Norm} = \sqrt{(x_t - x_{t-1})^2 + (y_t - y_{t-1})^2 + (z_t - z_{t-1})^2}. \tag{4}$$

This formula determines the change in intensity of the accelerometer's signals from one time frame to the next, illuminating the rate of change in motion. This comparison between the instantaneous norm and the difference norm aids in understanding both the magnitude and the variation of movement, facilitating the differentiation between fall and no-fall events.

## RESULTS

Our primary goal is to provide a valuable resource for research and development in advanced fall detection technologies. In this context, Table 3 offers a summary of the different types of falls, alongside corresponding measurements obtained from various sensors. Each column details the sensor readings in specific units: *radar* measurements in meters (m), *accelerometer* readings in meters per second squared (m/s²), *thermal sensor* data in degrees Celsius (°C), and *LIDAR* measurements also in meters (m). Each cell not only reports a mean value but also the standard deviation in parentheses, providing a detailed and quantitative perspective of each fall event.

Beyond these basic values, we have employed a set of additional metrics to more deeply characterize the captured data. These metrics include the instantaneous norm and the

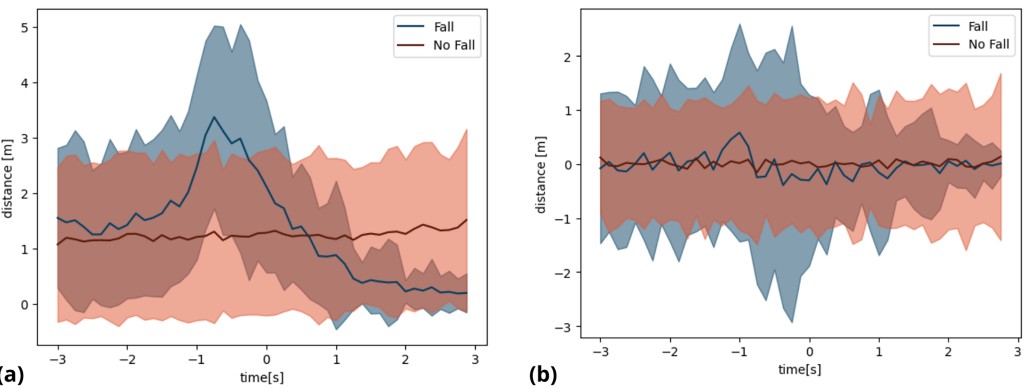

**Figure 5** Comparative analysis of radar signal metric (A) instantaneous norm of radar signals; (B) norm differences of radar signals between consecutive frames.

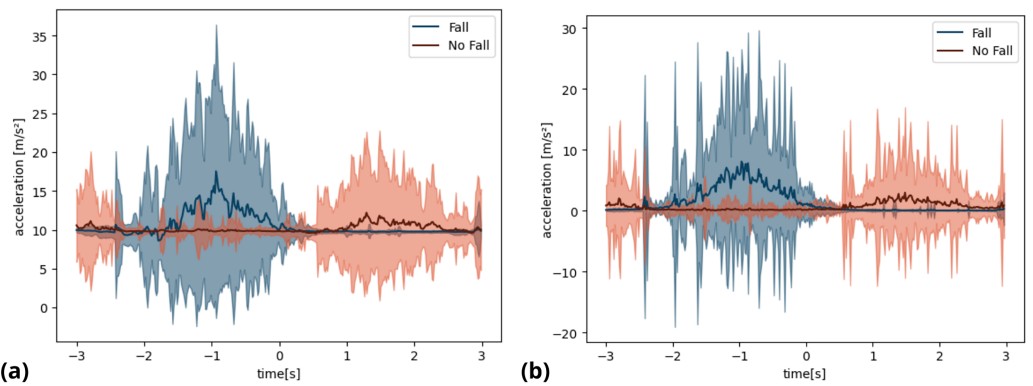

**Figure 6** Comparative analysis of accelerometer signal metrics: (A) instantaneous norm of accelerometer signals; (B) norm differences of accelerometer signals between consecutive frames.

difference in norms between consecutive frames, allowing for a finer assessment of the significant differences between fall and non-fall events. These evaluations are visually represented in Figs. 5, 6, 7 and 8 which complement and enrich our understanding of the interaction between different types of data and their relevance for practical applications in fall detection.

Our study applies two distinct metrics to characterize fall and no-fall events across four different sensor modalities: radar, accelerometer, thermal camera, and LIDAR. For each sensor type, we examine the instantaneous norm and the difference in norms between consecutive frames. Since our dataset includes ten falls across ten subjects, we present the mean of each metric accompanied by a dispersion range represented by a two-standard deviation interval around the mean.

The radar sensor data reveals significant differences between fall and no-fall events, particularly when examining the instantaneous norm, as shown in Fig. 5. This distinction is due to the Doppler effect, which captures the speed of moving objects and makes

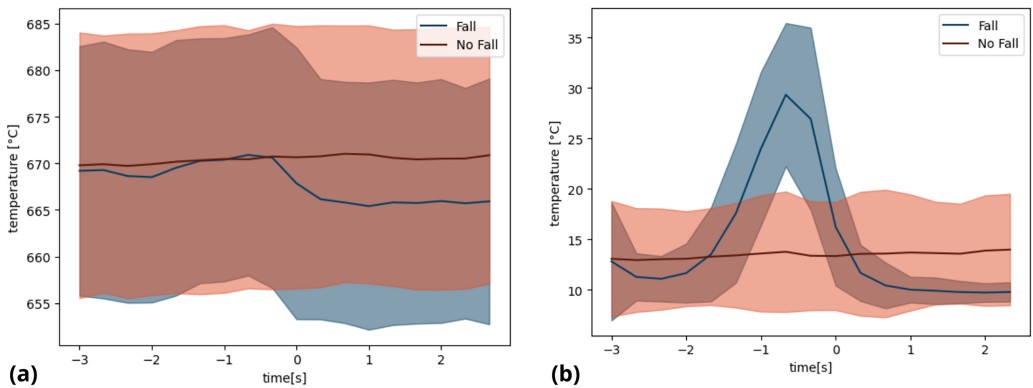

**Figure 7** Comparative analysis of thermal camera data metrics: (A) instantaneous norm of thermal data; (B) norm differences of thermal data between consecutive frames.

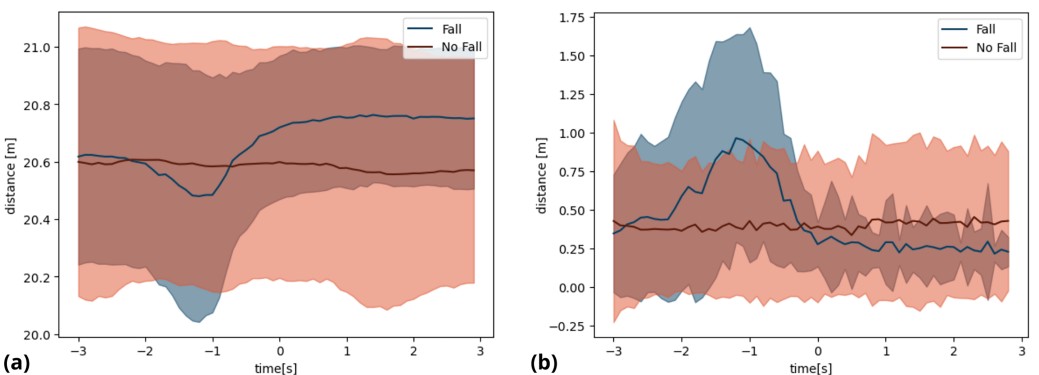

**Figure 8** Comparative analysis of LIDAR data metrics: (A) instantaneous norm of LIDAR; (B) norm differences of LIDAR data between consecutive frames.

falls distinguishable in terms of signal characteristics. In contrast, the difference metric, while still useful, shows a more subtle differentiation between the two event types. This statement suggests that although radar can clearly detect the immediate impact of a fall, the subsequent change is less noticeable.

Figure 6 shows the accelerometer data, which indicates a clear distinction between fall and no-fall events for both metrics. However, the first metric, representing the instantaneous norm, highlights the difference more clearly. This metric effectively captures the essence of the acceleration changes associated with a fall, with signals of significantly higher amplitude. This significant difference can be attributed to the accelerometer's inherent sensitivity to changes in motion, which makes the initial impact and subsequent movements more apparent.

Figure 7 presents the thermal camera data analysis, which shows a clear difference only in the second metric between fall and no-fall events. This is due to the fact that changes in the subject's position within the camera's field of view do not affect the overall intensity captured in the thermal images. As a result, the global intensity remains relatively stable

across both types of events, leading to minimal variation in the instantaneous metric. Therefore, the metric that focuses on differences between frames is particularly effective in highlighting falls because it captures the spatial displacement associated with such events.

The LIDAR sensor, depicted in Fig. 8, produces results that are comparable to those of the thermal camera. However, the difference metric is more effective in distinguishing falls from other activities. This metric highlights the spatial changes that are characteristic of a fall, which are less apparent when considering only the instantaneous norm. This passage highlights the importance of analyzing frame-to-frame variations in order to identify falls using the spatially dense data supplied by LIDAR.

## DISCUSSION

This study introduces a novel dataset that documents both fall and no-fall incidents, utilizing an array of four sensor types: LIDAR, thermal imaging, accelerometers, and radar. These sensors capture a variety of physical attributes, offering a detailed and multifaceted perspective on the dynamics of falls. The dataset facilitates the investigation of complex fall patterns and behaviors, which were previously challenging to discern, thus enhancing the development and fine-tuning of fall detection algorithms. While the dataset's scope—covering 10 individuals and 10 fall types—may seem restricted and its simulated nature potentially less valuable, it is crucial to emphasize that the goal of this project is to create a synchronized multisensor dataset for crafting more sophisticated data fusion solutions. Beginning with this foundational dataset allows us to establish initial performance benchmarks and delve into the intricacies of fall detection *via* sensor data fusion. This starting point also offers a chance to learn from simulated falls, preparing the groundwork for future studies involving more authentic fall scenarios. Looking ahead, we intend to broaden this initial dataset by including a wider range of scenarios and enlarging the participant pool. This expansion will build on the established baseline, progressively refining the reliability and practicality of our algorithms. The dataset serves the increasing need for synchronized multisensory datasets critical for testing and advancing such systems. It is extensive, including synchronized data from multiple sensors, and lays a solid foundation for developing algorithms capable of accurately detecting and predicting falls.

For our study, we used specific metrics to characterize the data derived from fall and no-fall events, which provided valuable insights. The analysis revealed that the instantaneous norm metric was more effective in distinguishing between fall and no-fall events for the radar and accelerometer sensors. This distinction can be attributed to the inherent properties of these sensors. They measure changes in velocity (Doppler effect) and acceleration, respectively, which are pronounced during fall events. In contrast, thermal cameras and LIDAR sensors were more effectively differentiated by the difference metric. A fall captured by an infrared camera does not necessarily induce a change in the instantaneous metric, such as a global intensity shift of the pixels. However, the difference between consecutive frames can reveal the distinction between fall and non-fall events. The variation in metric effectiveness across different sensor types highlights the complexity involved in accurately detecting falls and underscores the importance of selecting appropriate metrics to improve fall detection algorithm performance. The

dataset used in this study may present a limitation, as simulated falls can differ from real falls. Authentic falls are typically involuntary, with complex dynamics and compensatory movements that reduce impact. In contrast, simulated falls follow repetitive, specific patterns that lack this spontaneity, potentially reducing the accuracy of models trained exclusively on such data when applied in real-world environments (*Casilari & Silva, 2022*; *Bagalà et al., 2012*).

For future work, it would be beneficial to expand the comparative analysis between fall and no-fall events by including a wider range of variables and more complex scenarios. Specifically, incorporating scenarios that simulate everyday activities in more challenging and realistic settings could significantly improve the detection systems, making them more applicable to real-world environments. Also, we plan to include the continuous recording of an experimental subject performing daily activities over a prolonged period, which will provide a richer array of realistic signals to further enhance the model's robustness. We also plan to incorporate additional smartphone sensors, such as the gyroscope and magnetometer, to capture more detailed motion data and improve fall characterization. Furthermore, the use of next-generation sensors, which are more sensitive and compact, offers a promising way to improve the system. Adding technologies such as WiFi and Kinect to the sensor suite could further improve fall detection and daily activity monitoring systems, making them more accurate and effective in supporting the elderly. On the other hand, our dataset is slightly unbalanced, with a ratio of 1 to 3 between fall and no-fall events. In future work, we plan to address this imbalance using techniques such as oversampling or undersampling to generate synthetic data for the minority class. Furthermore, identifying movement patterns during a fall is a complex challenge, and to minimize the risk of injury we work with young people. However, this limits the representation of different levels of mobility that may be present in older people and partly restricts the applicability of the system. Despite the efforts made in this research, future consideration should be given to including a variety of mobility levels in order to ensure the effectiveness of the system for a broader spectrum of users. Another important aspect for future research is the improvement of wearable devices used in fall detection. Since these devices are often subjected to torsion and other forces during daily activities, it is crucial that they are both flexible and durable. A promising option is the application of flexible electronics technologies, as discussed in the research (*Gao et al., 2024*). This study suggests that the integration of flexible materials can significantly enhance the adaptability and comfort of wearable sensors, which is essential to ensure their functionality and acceptance by users in real environments. Additionally, another future improvement to consider is the integration of various sensors into a single device that allows for real-time data reading and analysis. Currently, data acquisition is performed separately for each sensor, which can discourage adoption due to the dispersion of the devices and their complex placement. An integrated device could significantly simplify usage and notably improve the user experience.

According to our study framework, our study focused on the identification of fall patterns, without considering movements after the fall event. Considering post-fall movements could enrich the analysis, providing valuable information for immediate intervention decisions and in the future for the development of specific rehabilitation

strategies. Given these considerations, part of the future focus of the study will be to include, within the study, post-fall event patterns.

Our main objective is focused on the technical aspects such as sensors selection and measure of signals for the different output, but we recognize the importance of real world validation for the demonstration of the practical usefulness of our methods in fall detection scenarios. Although this study does not include validation in real-world implementations, we consider this aspect as crucial for future research. We recognize the necessity of real-world validation, for example field trials in various environments (*Broadley et al., 2018*), where the falls are likely to occur. By addressing this aspect, we want to enhance the applicability and reliability of our method for real-world fall detection applications.

### Funding

This work was supported by the FONDECYT Regular project 1201787 "Multimodal Machine Learning approach for detecting pathological activity patterns in elderlies" and the FOVI220145 project "International collaboration program for research and development of intelligent environments." The work was further supported by ANID - MILENIO - NCS2021_013. There was no additional external funding received for this study. The funders had no role in study design, data collection and analysis, decision to publish, or preparation of the manuscript.

### Grant Disclosures

The following grant information was disclosed by the authors:
The FONDECYT Regular project 1201787 "Multimodal Machine Learning approach for detecting pathological activity patterns in elderlies".
The FOVI220145 project "International collaboration program for research and development of intelligent environments".
ANID - MILENIO - NCS2021_013.

### Competing Interests

The authors declare there are no competing interests.

### Author Contributions

- Carla Taramasco conceived and designed the experiments, authored or reviewed drafts of the article, and approved the final draft.
- Miguel Pineiro conceived and designed the experiments, prepared figures and/or tables, authored or reviewed drafts of the article, and approved the final draft.
- Pablo Ormeño-Arriagada analyzed the data, prepared figures and/or tables, and approved the final draft.
- Diego Robles performed the experiments, authored or reviewed drafts of the article, and approved the final draft.
- David Araya conceived and designed the experiments, authored or reviewed drafts of the article, and approved the final draft.

## Human Ethics

The following information was supplied relating to ethical approvals (i.e., approving body and any reference numbers):

This study was conducted with the approval of the Ethics Committee of Universidad Andrés Bello, under Approval Act number 032/2023.

## Data Availability

The data is available at OSF: Pineiro, Miguel. 2024. ''Multi-Sensor Fall Detection.'' OSF. November 4. doi:10.17605/OSF.IO/YJGDV.

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
