# Peer review of "Multimodal dataset for sensor fusion in fall detection"

_PeerJ, doi:10.7717/peerj.19004_

## Round 0.1 · original submission · Major Revisions

Please consider the reviewers' comments.

Reviewer 1 ·

Basic reporting

The dataset and its description should be improved.
Refer to the additional comments for further information.

Experimental design

The number of ADLs and the number participants in the tests are too low.
A more detailed description of the testbed is required
See the additional comments.

Validity of the findings

The dataset could be of interest if improved.
The preliminary analysis of the data is poor.
See the additional comments.

Additional comments

The developed dataset is of great interest and is supported by an acceptable review of the existing datasets (although there certain public databases that are neglected). The idea of generating and using multisensory databases to detect falls is not new, although it is true that there are not many public databases that simultaneously offer data from wearable sensors (inertial sensors) and contextual sensors (cameras, infrared sensors). In this sense, the collected database offers some novelty.

Since there are many inertial and contextual fall databases, in my opinion, Table 1 should be limited to those databases that combine both systems. Some of the databases present in that table (such as SisFall and UMAFall) only offer inertial data. It is true that they include some videos of how the falls and ADLs were generated, but they are merely illustrative clips. In those databases, most of the movements were characterized solely by inertial signals, without any added video signal. In the case of MobiFall, I believe that it does not even include any videos, and all the signals offered were captured by portable inertial sensors.

Both contextual and wearable systems present a number of disadvantages that should be mentioned. For example, camera-based systems can be vulnerable to changes in the monitored environment. For instance, what happens if the user moves a piece of furniture? Furthermore, the installation, maintenance, and adjustment of these systems by an expert can involve significant costs. I understand that this is a secondary aspect, but it should at least be mentioned.

The use of databases with simulated falls by volunteers to evaluate fall detection systems is extremely controversial and is being increasingly contested. A fall is inherently an unplanned, unstructured, involuntary act that results in highly variable and complex dynamics, as the injured person performs compensatory movements to minimize damage. These compensatory movements (which reduce the violence of the impact against the flor) are not performed during ‘fake’ falls mimicked by trained participants who have received prior instructions on how to fall and who follow a limited set of very specific and repetitive patterns. These emulated patterns can be easily discriminated from other typical ADLs by artificial intelligence methods. Some articles have shown that the dynamics of real falls may resemble certain everyday movements (e.g., climbing stairs or jumping) more than fake falls. This fact is not discussed by the authors.

Fall detection systems (FDSs) must discriminate falls from other types of activities of daily living (ADLs). Therefore, it is important that the dataset includes not only many types of falls (even if they are simulated) but also a varied number of ADLs. In this case, the database is quite poor as it only incorporates one very basic ADL: walking. Distinguishing a fall from the simple act of walking is very easy for almost any detector (whether threshold-based or using artificial intelligence).

However, there are more complex movements that can be more easily confused with falls: bending down to pick up an object, jumping, lying down/getting up from a bed, sitting down/getting up from a chair, crawling to look for something, descending from a ladder, opening/closing a door, etc. The databases mentioned and reviewed in the literature incorporate many more ADLs. I consider it necessary to include movements of a different nature (besides walking).

Another option is to record an experimental subject over a long period while they go about their daily life. This way, a wide array of realistic signals can be obtained.

The presence of mattresses and other cushioning elements also reduces the realism of the mobility patterns, and I am unsure to what extent they alter the recordings from the cameras compared to what would be measured in the case of a real fall."

The measurements of the inertial signals are taken on the back using a smartphone. The authors should indicate at what height (e.g., above which vertebra) the phone is placed. In any case, considering the practical application of a potential detector, the back is not the best position to place a sensor for ergonomic reasons. Most datasets include measurements at the waist, the thigh (pocket) or, at most, at the chest. The authors do not justify the choice of the back.

The description of the testbed should be improved: the model of the smartphone and the manufacturer of the accelerometer it carries are not specified. The range and resolution of the accelerometer are also not indicated.

A problem of using a smartphone instead of a specific inertial sensor (such as Shimmer tags or similar sensing motes) is that it is difficult to accurately establish the sampling frequency. The authors do not indicate or justify what sampling frequency they use.

The vast majority of smartphones also include a gyroscope and a magnetometer. Why has the inclusion of these measurements not been considered in the dataset?

Are the clocks of the four sensors synchronized? If so, how is this synchronization achieved? It is not completely clear whether it is done through mere visual inspection of the video sequences. How is the moment corresponding to the fall determined in the acceleration measurements? Is it identified by the peak caused by the impact?

Most datasets use a larger number of experimental subjects (as shown in Table 1). Ten subjects is too few. Additionally, their personal characteristics (weight, age, gender, height) should be indicated.

In my opinion, the article should focus on providing a detailed description of the dataset. The authors offer a preliminary analysis of the traces by selecting certain features, which they do not justify sufficiently. In this regard, especially concerning wearable devices, there is a considerable amount of literature discussing the selection of such features. The choice of a window of 4 seconds around the moment of the fall is reasonable but should be justified further. There is quite a bit of literature on this aspect (and smaller windows are usually preferred, as most falls do not last that long).

In any case, comparing the dynamics of falls with a single type of ADL (walking) does not make much sense. A much broader typology of ADLs should be considered.

Table 2: it does not make much sense to express the mean value and the standard deviation of the acceleration magnitude.

Is the URL where the generated dataset is available indicated?

Other aspects: It would be interesting to include a photograph of the setup, at the location where the tests were conducted. Figure 3(b): Is the magnitude of the acceleration vector actually represented? Or is it the value of one of the components? In the database, which column represents each axis of the accelerometer? Page 9. Typo: “These advanced evaluations are visually represented in Figures ??, ??, and ??."

Cite this review as

Reviewer 2 ·

Basic reporting

This paper presents a new dataset for fall detection combining several sensor modalities. The reporting is clear and concise, and the authors should be commended for providing a valuable contribution in the form of the dataset.

Please find my requests and suggestions below.

=========================
LITERATURE
=========================
In the introduction, a good summary of recent publications on fall detection has been provided; however, a couple of substantial areas have been omitted. Firstly, the use of robots in fall detection: With the growing capabilities of care robots, this is likely to be an important platform-class for future applications of fall detection; particularly as they hold the potential to not only detect, but also respond to falls. I would ask that the authors cite some of the recent and seminal papers in the area of using robots for fall detection, such as:

Fischinger D, et al. (2016) Hobbit, a care robot supporting independent living at home: First prototype and lessons learned. Robotics and autonomous systems 75: 60-78.

Wei H, et al. (2024) A Novel Fall Detection System Using the AI-Enabled EUREKA Humanoid Robot. Advances in Intelligent Manufacturing and Robotics. Springer Nature. doi:10.1007/978-981-99-8498-5_41

Elwaly A, et al. (2024) New Eldercare Robot with Path-Planning and Fall-Detection Capabilities. Applied Sciences. 14(6):2374.

Along with interventions to detect and respond to age-related injuries, it is important to acknowledge work in other areas to address the problem of age-related frailty and falls, including the growing call for approaches to detect and treat age-related pathologies before they lead to frailty:

Calimport SRG, et al. (2019) To help aging populations, classify organismal senescence. Science, 366(6465). doi:10.1126/science.aay7319

=========================
DATA
=========================
Please include the link to the dataset in the main text, along with the repository name and accession number.

=========================
STYLISTIC COMMENTS
=========================

Please address the following:
28: Use the multiplication symbol rather than x.
122-124: Please do not have a single sentence paragraph. Incorporate into the surrounding paragraphs.
155: Use left and right quotation marks, not just right.
172: Use multiplication symbol.
174: Use multiplication symbol.
175: Use multiplication symbol.
196: Use of an exclamation point is inappropriate here.
217: Please do not put units in italics (m/s2).
219: Use the multiplication symbol rather than x.
276: Use left and right quotation marks, not just right.
284: Replace chapter with section. This first sentence also does not state anything of value. Please consider removing.
291: See previous comment on not italicising units.
297: Figure numbers missing.
386: Indent missing.

LiDAR, LIDAR, and lidar are all used in the paper. Likewise, there is a mix of RADAR, Radar, and radar. Please choose a single style and use it consistently.

Table 2: Please specify in the caption what the main values and those in parentheses represent (this is in the main text, but it should also be with the table). Note also that m/s2 should not be italicised, and there is a missing right parenthesis on the last line. Finally, please include a line under the header, and beneath the final row, as in Table 1.

Experimental design

No comments on experimental design.

Validity of the findings

The demographics / composition of the participants used for data collection is important to know – please make this clear in the text. A fall by a young underweight female, is going to have very different dynamics to an elderly obese male. If participants are too dissimilar from the true population, is possible that the training data might not generalise. In the interest of precision, please discuss the demographics and point this potential shortcoming.

Additional comments

With the exception of the stylistic comments in Section 1, I consider all other requests to be major. Please correct these before resubmission.

Cite this review as

Reviewer 3 ·

Basic reporting

All comments have been added in detail to the last section.

Experimental design

All comments have been added in detail to the last section.

Validity of the findings

All comments have been added in detail to the last section.

Additional comments

Review Report for PeerJ
(Multimodal dataset for sensor fusion in fall detection)

1. Within the scope of the study, both sensor fusion algorithms can be developed better and advanced multisensor fall detection algorithms can be evaluated by a multisensor dataset.

2. The importance of the subject is clearly mentioned in the introduction section. However, the differences of this study from the literature and its main contributions to the literature should be stated more clearly and in detail in the end of this section.

3. In the Related works section, fall detection and sufficient literature on the subject are mentioned. In addition, it is observed that the public dataset comparison in Table-1 is also very suitable and sufficient for the study.

4. Although it is important to obtain an ethical committee for the study, when the data description, block diagram of the multisensor system setup, feature extraction sections and dataset segmentation sections are examined in detail, it is understood that the quality of the study is clearly revealed.

5. It is observed that the results and metric types obtained are sufficient when compared to the literature.

As a result, the study can make a significant contribution to the literature, but the above sections should be examined carefully.

Cite this review as

---

## Round 0.2 · Major Revisions

Dear authors, a deep discussion on reviewer 1 comments is needed.

Reviewer 1 ·

Basic reporting

Authors have remarkably modified the text of the paper and have commented in more detail some aspects related to the description of the dataset and the testbed where it was generated. However, they have not addressed some of my most relevant comments: the dataset, which has neither been modified nor extended, just includes a single type of ADL (walking). It has no sense to identify the difference in the dynamics of falls and ADLs if the only ADL under consideration is walking (which is extremely different from falling).
Some suggestions to improve the dataset are just considered as future work (such as long-term monitoring, considering more experimental subjects or including the measurements of other sensors such as gyroscope, which are already integrated in the smartphone).

Experimental design

see basic reporting

Validity of the findings

see basic reporting

Additional comments

see basic reporting

Cite this review as

Reviewer 2 ·

Basic reporting

No comment

Experimental design

No comment

Validity of the findings

No comment

Additional comments

All of the comments from the first round of reviews have been sufficiently addressed, and the updated paper is greatly improved.

I am happy to recommend this for publication.

Cite this review as

---

## Round 0.3 · accepted · Accept

The revised manuscript was sent to one of the original reviewers, who initially requested major revisions. However, this reviewer declined to review the updated version. Consequently, I undertook a thorough assessment of the revision myself. I am pleased to report that you have successfully addressed all the reviewers' comments and have made the necessary improvements to enhance the manuscript. With these revisions, I am satisfied with the current version and deem it ready for publication.